# Antibiotic prescribing for lower UTI in elderly patients in primary care and risk of bloodstream infection: A cohort study using electronic health records in England

**Laura Shallcross**[1☯]*, **Patrick Rockenschaub**[1☯], **Ruth Blackburn**[1], **Irwin Nazareth**[2], **Nick Freemantle**[3], **Andrew Hayward**[4]

1 Institute of Health Informatics, University College London, London, United Kingdom, 2 Department of Primary Care and Population Health, University College London, London, United Kingdom, 3 Institute of Clinical Trials and Methodology, University College London, London, United Kingdom, 4 Institute of Epidemiology & Healthcare, University College London, London, United Kingdom

☯ These authors contributed equally to this work.
* l.shallcross@ucl.ac.uk

**Data Availability Statement:** Clinical Practice Research Datalink (CPRD), Hospital Episode Statistics (HES) and Office for National Statistics

## Abstract

### Background

Research has questioned the safety of delaying or withholding antibiotics for suspected urinary tract infection (UTI) in older patients. We evaluated the association between antibiotic treatment for lower UTI and risk of bloodstream infection (BSI) in adults aged ≥65 years in primary care.

### Methods and findings

We analyzed primary care records from patients aged ≥65 years in England with community-onset UTI using the Clinical Practice Research Datalink (2007–2015) linked to Hospital Episode Statistics and census data. The primary outcome was BSI within 60 days, comparing patients treated immediately with antibiotics and those not treated immediately. Crude and adjusted associations between exposure and outcome were estimated using generalized estimating equations.

A total of 147,334 patients were included representing 280,462 episodes of lower UTI. BSI occurred in 0.4% (1,025/244,963) of UTI episodes with immediate antibiotics versus 0.6% (228/35,499) of episodes without immediate antibiotics. After adjusting for patient demographics, year of consultation, comorbidities, smoking status, recent hospitalizations, recent accident and emergency (A&E) attendances, recent antibiotic prescribing, and home visits, the odds of BSI were equivalent in patients who were not treated with antibiotics immediately and those who were treated on the date of their UTI consultation (adjusted odds ratio [aOR] 1.13, 95% CI 0.97–1.32, p-value = 0.105). Delaying or withholding antibiotics was associated with increased odds of death in the subsequent 60 days (aOR 1.17, 95% CI 1.09–1.26, p-value < 0.001), but there was limited evidence that increased deaths were attributable to urinary-source BSI.

(ONS) data cannot be directly shared by the researchers but are available directly from CPRD and NHS Digital subject to standard conditions. All statistical code is available from https://github.com/prockenschaub/CPRD_UTI_sepsis_elderly.

**Funding:** This work was supported by the Economic and Social Research Council (ES/P008321/1) and by Health Data Research UK, an initiative funded by UK Research and Innovation, Department of Health and Social Care (England) and the devolved administrations, and leading medical research charities. LS was funded by a National Institute for Health Research (NIHR) Clinician Scientist award (CS-2016-007) for this research project. RB was supported by a UKRI Innovation Fellowship funded by the Medical Research Council (Grant No: MR/S003797/1), AH was funded by an NIHR Senior Investigator award for this project. The funders had no role in study design, data collection and analysis, decision to publish, or preparation of the manuscript. The corresponding author had full access to all the data in the study and had final responsibility for the decision to submit for publication.

**Competing interests:** The authors have declared that no competing interests exist.

**Abbreviations:** A&E, accident and emergency; aOR, adjusted odds ratio; BSI, bloodstream infection; CCI, Charlson Comorbidity Index; CI, confidence interval; CPRD, Clinical Practice Research Datalink; EHR, electronic health record; GP, general practitioner; HES, Hospital Episode Statistics; ICD-10, International Classification of Diseases 10th revision; IMD, Index of Multiple Deprivation; MHRA, Medicines and Healthcare products Regulatory Agency; NHS, UK National Health Service; NIHR, UK National Institute for Health Research; OR, odds ratio; Q, quintile; UTI, urinary tract infection.

Limitations include overlap between the categories of immediate and delayed antibiotic prescribing, residual confounding underlying differences between patients who were/were not treated with antibiotics, and lack of microbiological diagnosis for BSI.

## Conclusions

In this study, we observed that delaying or withholding antibiotics in older adults with suspected UTI did not increase patients' risk of BSI, in contrast with a previous study that analyzed the same dataset, but mortality was increased. Our findings highlight uncertainty around the risks of delaying or withholding antibiotic treatment, which is exacerbated by systematic differences between patients who were and were not treated immediately with antibiotics. Overall, our findings emphasize the need for improved diagnostic/risk prediction strategies to guide antibiotic prescribing for suspected UTI in older adults.

## Author summary

### Why was this study done?

- Urinary tract infections (UTI) are common in older adults and, alongside respiratory infections, account for the majority of antibiotics prescribed in primary care

- Antibiotics are often prescribed inappropriately for UTI in the elderly, but the need to reduce prescribing must be balanced against the risk of increasing rare but severe outcomes, such as bloodstream infection, if antibiotic treatment is delayed

- A recent study in patients aged >65 years found that those who did not receive immediate antibiotic treatment for UTI were more likely to develop bloodstream infection

### What did the researchers do and find?

- We reanalyzed the relationship between the timing of antibiotic prescribing for UTI and subsequent risk of bloodstream infection (BSI) using the same dataset

- We did not find evidence to suggest that not immediately prescribing antibiotics for UTI increased a patient's risk of bloodstream infection, but we did find some evidence of increased mortality.

- Women were less likely to develop BSI compared with men (adjusted odds ratio [aOR] 0.49, 95% confidence interval [CI] 0.43–0.55, $p$-value < 0.001). Increasing age (aOR 1.22, 95% CI 1.18–1.27 per 5 years, $p$-value < 0.001) and social deprivation (Q5 versus Q1: aOR 1.45; 95% CI 1.19–1.76, $p$-value < 0.001) were also independently associated with BSI.

- Systematic differences between patients who were/were not treated immediately with antibiotics (residual confounding) remains a potential explanation for our findings in relation to mortality.

**What do these findings mean?**

- This population-based study highlights uncertainty around whether delaying antibiotics in older adults with suspected UTI increases their risk of adverse outcomes.

- The reasons for the systematic differences identified between patients who were and were not treated immediately with antibiotics warrants further study.

- Adverse consequences of antibiotic treatment in this population and the public health need to tackle antibiotic resistance highlight the need for novel diagnostic and/or risk prediction strategies to guide antibiotic prescribing decisions for suspected UTI.

## Introduction

Urinary tract infections (UTIs) are common in older adults in both primary and secondary care [1], with *Escherichia coli* as the causative pathogen in 70%–95% of cases [2]. The clinical spectrum of UTI ranges from mild urinary symptoms to urosepsis, but the rate of *E. coli* bloodstream infection is highest in the oldest age groups (758.5/100,000 in ≥85 years versus 53.4/100,000 in 45–64-year-olds) [3].

Identifying cases of UTI can be challenging, particularly in the elderly, who often present with atypical signs and symptoms of infection [4]. Diagnostic uncertainty is compounded by the increased prevalence of asymptomatic bacteriuria in older adults (>20% in women aged ≥65 years compared with 5% of younger women) [5,6] and widespread use of urine dipstick testing across healthcare settings, despite its poor positive predictive value for bacteriuria [7]. Older patients are also at disproportionate risk of toxicity from antibiotics, as well as complications such as *Clostridium difficile* infection [8], adding to the complexity of management decisions.

UTI is the second commonest reason for antibiotics to be prescribed in primary care. An estimated 40%–50% of antibiotic prescriptions for UTI are estimated to be inappropriate [9], although the degree of inappropriate prescribing varies widely across settings and countries [10]. A wide range of national initiatives aiming to tackle inappropriate prescribing have reduced total prescribing by 13.2% between 2013 and 2017 [11], mainly by reducing prescribing for respiratory tract infections. This has reduced total prescribing of broad-spectrum antibiotics, even in elderly populations [12]. However, rates of gram-negative bloodstream infections (BSIs) continue to rise [11], and although it is anticipated that reductions in prescribing will have a beneficial impact on rates of antibiotic resistance and *C. difficile* infection, this has to be balanced against the risk of increasing rare but severe outcomes such as BSI.

The safety of delaying or withholding antibiotic treatment for suspected UTI in older adults in primary care was recently investigated in an electronic health record (EHR) study by Gharbi and colleagues. This study reported a 7–8-fold increase in the odds of BSI in the 60 days following consultation if antibiotic treatment was delayed or withheld by comparison with patients who were treated immediately (i.e., on the date of their first UTI consultation) [13]. Delaying or withholding antibiotics was also associated with a statistically significant increase in 60-day mortality. To the best of our knowledge, Gharbi and colleagues are the first to address this important research question, and their findings are therefore likely to have a significant influence on policy and clinical practice, for example, by reducing general practitioners' (GPs) willingness to consider the use of potentially beneficial strategies such as delayed

prescribing. However, a number of research groups have strongly questioned the validity of these findings [14], highlighting methodological concerns around the definition of UTI episodes and the comparability and definitions of the different antibiotic treatment groups.

GPs require robust evidence on which to base empirical prescribing decisions, and in the absence of randomized controlled trials, observational studies using large-scale EHRs can help to address this evidence-gap. We therefore attempted to replicate the findings reported by Gharbi and colleagues by analyzing the same dataset and undertaking a range of sensitivity analyses to test the robustness of our findings. We addressed the following research question: In a population aged ≥65 years who consult primary care for suspected lower UTI, are patients who are not treated with antibiotics immediately at increased risk of BSI in the 60 days following consultation, compared with patients who were treated with an antibiotic on the date of their consultation?

## Methods

### Database and study population

The Clinical Practice Research Datalink (CPRD) database is a nationally representative database of primary care consultations in the United Kingdom [15]. Data in CPRD are collected anonymously from practice management systems of 674 practices and include demographic information, medical tests, diagnoses, and prescriptions. Diagnoses are entered directly by clinicians using Read codes, the main medical coding terminology in UK primary care [16]. A subset of consenting English patients and practices (75% of English practices, 58% of all practices) are further linked to data on hospital admissions and visits to the Emergency Department from the Hospital Episode Statistics (HES) and census data from the Office for National Statistics (ONS).

We included all patients in the CPRD-HES-ONS linked data aged 65 years or more between April 1, 2007, and March 31, 2015. Data were required to fulfil basic quality standards [15], and patients entered the cohort at the latest of the practice's up-to-standard date, 1 year of continuous registration with the practice, their 65th birthday, or April 1, 2007. Patients left the cohort either on their date of death or 60 days before the earliest of the practice's last collection date, their transfer-out date, or March 31, 2015. All included patients had a minimum of 60 days follow-up, with the exception of patients who died, because excluding these individuals would bias our results.

### Ethical approval

This study was conducted as part of the Preserving Antibiotics through Safe Stewardship (PASS) project [17]. Access to CPRD data within PASS was approved by the Medicines and Healthcare products Regulatory Agency (MHRA UK) Independent Scientific Advisory Committee (ISAC-Nr.: 17 048), under Section 251 (UK National Health Service [NHS] Social Care Act 2006). Individual patient consent was not required for this observational study of anonymized data.

### Definition of UTI episodes

The study population comprised patients who consulted for a new episode of lower UTI that originated in the community. Primary care consultation for community-onset lower UTI was identified from the primary care record based on Read codes using previously published code-lists [13] (S1 Table) and supplemented with data from the linked hospital record based on International Classification of Diseases 10th revision (ICD-10) codes to exclude cases that

originated in hospital. A major challenge in the analysis of routine data is distinguishing between new and ongoing episodes of infection and differentiating between community-onset and hospital-onset infections. For example, a patient could consult primary care for urinary symptoms twice in a 3-month period and depending on the interval between events, this might be classified as 1 or 2 episodes of UTI. Similarly, a patient may be diagnosed with UTI in hospital but consult primary care for urinary symptoms 2 weeks later following discharge. Here, the primary care event is likely to represent continuation of a UTI that originated in hospital.

For these reasons, we applied a strict definition of UTI to restrict our analysis to community-onset cases of lower UTI and to differentiate between new and ongoing UTI episodes. For each patient, the first episode of infection was defined as the date of the earliest observed UTI code in primary and/or secondary care (Fig 1). Like Gharbi and colleagues, we considered any

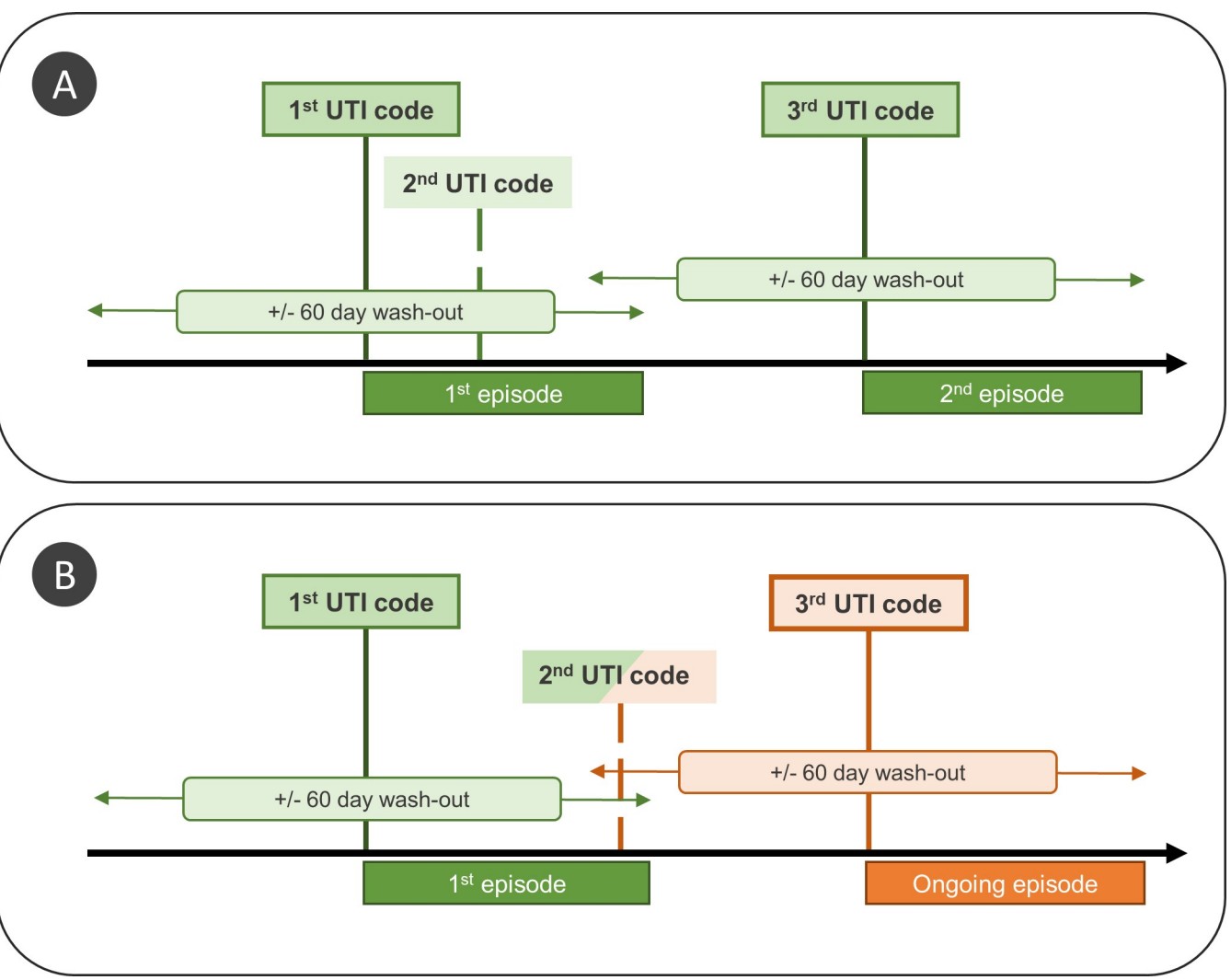

**Fig 1. Classification of UTI episodes for 2 scenarios for a patient with 3 records of UTI, which are identical except for the timing of the second UTI code.** In both panels, the first UTI code marks the start of a new UTI episode (first episode). The second UTI code occurs within 60 days and is therefore considered to be part of the first episode. The third UTI code occurs more than 60 days after the start of the first episode and is classified as (A) a new episode (because the last evidence of UTI was recorded more than 60 days earlier); (B) an ongoing episode that is excluded from the analysis (because the last evidence of UTI, i.e., second UTI code, was recorded less than 60 days before and may therefore represent an ongoing episode of infection). UTI, urinary tract infection.

UTI codes recorded in the following 60 days to be part of the same episode. However, whereas Gharbi and colleagues used a fixed time period to define a UTI episode, we applied rules to differentiate between new and ongoing episodes of UTI, based on patterns of consultation for UTI recorded during the 60-day follow-up period (Fig 1A and Fig 1B).

We further excluded episodes in which the patient had any of the following outcomes recorded on the same day: hospital admission, accident and emergency (A&E) attendance, referral to specialist care, and death. Episodes were also excluded if the linked HES record showed that the patient was in hospital on the date that the episode was recorded in primary care.

## Exposure, outcomes, and covariates

We compared patients who were immediately treated with antibiotics defined as prescription of systemic antibiotics on the same day as the episode start date to patients who were not treated with antibiotics on the same day. In contrast to Gharbi and colleagues, we considered patients who were not prescribed antibiotics and those with a delayed prescription (i.e., antibiotics prescribed in the 7 days after—but not including—the day of initial consultation) as a single group, because delayed antibiotic prescribing is not well recorded in EHRs [18].

The primary outcome was BSI within 60 days of the episode start date recorded in the primary or secondary care record. Although the terms sepsis and BSI are not interchangeable, ICD-10 diagnostic codes usually record "sepsis" [19] rather than BSI, even in cases with a positive microbial culture of blood [20]. We have therefore interpreted an ICD-10 code for sepsis as evidence of BSI and used the term BSI throughout. Secondary outcomes were all-cause mortality within 60 days, admission to hospital for reasons unrelated to UTI or BSI within 60 days, and underlying cause of BSI. A 60-day follow-up for these outcomes was selected for purposes of comparison with the literature, notably Gharbi and colleagues. BSI was identified in primary care using Read codes and in secondary care using ICD-10 codes (which represent the primary and secondary reasons for admission) using published codelists [13]. ICD-10 codes for sepsis were further classified as urosepsis, sepsis of other infectious origin, and unspecified sepsis (S1 Text).

Explanatory variables included demographic characteristics: age at episode start, gender, quintile of socioeconomic status (Index of Multiple Deprivation [IMD] 2015), and practice region (South of England, London, East of England and Midlands, North of England and Yorkshire). We also evaluated risk factors for infection and healthcare utilization including Charlson Comorbidity Index (CCI), smoking status (nonsmoker, ex-smoker, current smoker), whether the index consultation was performed as a home visit, recent hospitalizations (discharge in prior 7 and 30 days, number of admissions in prior year, total number of days spent in hospital in prior year), recent A&E attendances (attendance in prior 30 days, number of attendances in prior year), and prescription of systemic antibiotics in primary care in prior 30 days. History of recurrent UTI was defined as an explicit code for recurrent UTI, a prescription of nitrofurantoin or trimethoprim for 28 days or more (prophylactic treatment), or 2 or more consultations for UTI within a year of episode start [13]. CCI and smoking status were calculated using all medical history in primary care before the episode start date. Patients without a smoking code were considered nonsmokers. Patients whose latest record indicated a nonsmoker but who had a previous record of smoking were classified as ex-smokers.

## Statistical analysis

We undertook a univariable analysis comparing patients with and without immediate antibiotic treatment for each included variable. Continuous variables were summarized using means

median and interquartile range (IQR), and categorical variables using absolute numbers and proportions. Wilcoxon rank-sum tests (continuous) and $\chi^2$ tests (categorical) were used to assess the difference between exposure groups. We tabulated diagnostic information relating to the underlying cause of BSI for each treatment group.

Crude associations (odds ratios [ORs]) between each included variable and BSI were estimated using generalized estimating equations (GEEs) with a logit link and an exchangeable correlation structure to account for multiple UTI episodes per patient. All count variables (CCI, number of admissions, number of days spent in hospital, and number of A&E attendances) were transformed using the square root before adding them to the GEEs. Huber-White sandwich estimators were used to calculate 95% confidence intervals (95% CI). A final multivariable adjusted model was fitted, including all predictors with a $p$-value $< 0.2$ in the univariable analysis. Based on reviewer comments, interactions between prescribing and age or gender were also considered. The number needed to be exposed (i.e., not treated with antibiotics) to harm (NNEH) was calculated from the final model using average risk difference to adjust for covariate imbalance [21]. The analysis was refitted in 200 bootstrapped samples to estimate 95% CIs for the NNEH.

The same approach was used for secondary outcomes. Sensitivity analysis was undertaken restricting the follow-up/wash-out periods to 30 days and only including the first UTI episode per patient. We also tested the sensitivity to residual confounding by performing propensity score analysis. A patient's prior likelihood to receive treatment was estimated using multivariable logistic regression (parametric) or generalized boosted regression (nonparametric), and 4 different adjusted results were obtained using each set of propensity scores with either matching or inverse probability weighting.

The analysis presented here was outlined prospectively in the protocol submitted to the MHRA Independent Scientific Advisory Committee for ethical approval (S1 Protocol). The definitions and methods were chosen to replicate the analysis performed by Gharbi and colleagues [13] as closely as possible and to address all concerns raised by researchers regarding the validity of those findings [14]. Analysis was performed using the statistical software R version 3.6.1 for Windows [22]. Generalized estimating equations were fitted using *geepack* (version 1.2–1), and propensity score analysis was performed using *MatchIt* (version 3.0.2) and *twang* (version 1.5). This study is reported as per the REporting of studies Conducted using Observational Routinely collected Data (RECORD) guideline (S1 RECORD Checklist). Code for all analyses can be found at https://github.com/prockenschaub/CPRD_UTI_sepsis_elderly.

## Results

Data were available for 850,794 patients aged ≥65 years corresponding to 3,706,722 patient-years at risk between April 1, 2007, and March 31, 2015 (Fig 2). The cohort included 147,334 patients with 280,462 distinct episodes of lower UTI, corresponding to 75.7 episodes per 1,000 patient-years at risk. UTI episodes mainly occurred in women 217,425/280,462 (77.5%).

Most UTI episodes (244,963/280,462; 87.3%) were treated with antibiotics immediately (Table 1), and at least 1 antibiotic prescription was recorded in the 7 days following consultation for 6411/35499 (2.3%) UTI episodes that were not treated immediately. Factors that were associated with delayed or withheld prescribing (versus immediate treatment) included male gender (40.9% versus 19.8%); antibiotic prescription in the previous 30 days (27.0% versus 18.2%), and GP home visits (9.6% versus 3.7%).

BSI was recorded in 1,025/244,963 (0.4%) UTI episodes with immediate antibiotic treatment and in 228/35,499 (0.6%) episodes that were not treated immediately (Table 1). The median number of days to diagnosis of BSI was shorter in patients who were not treated with

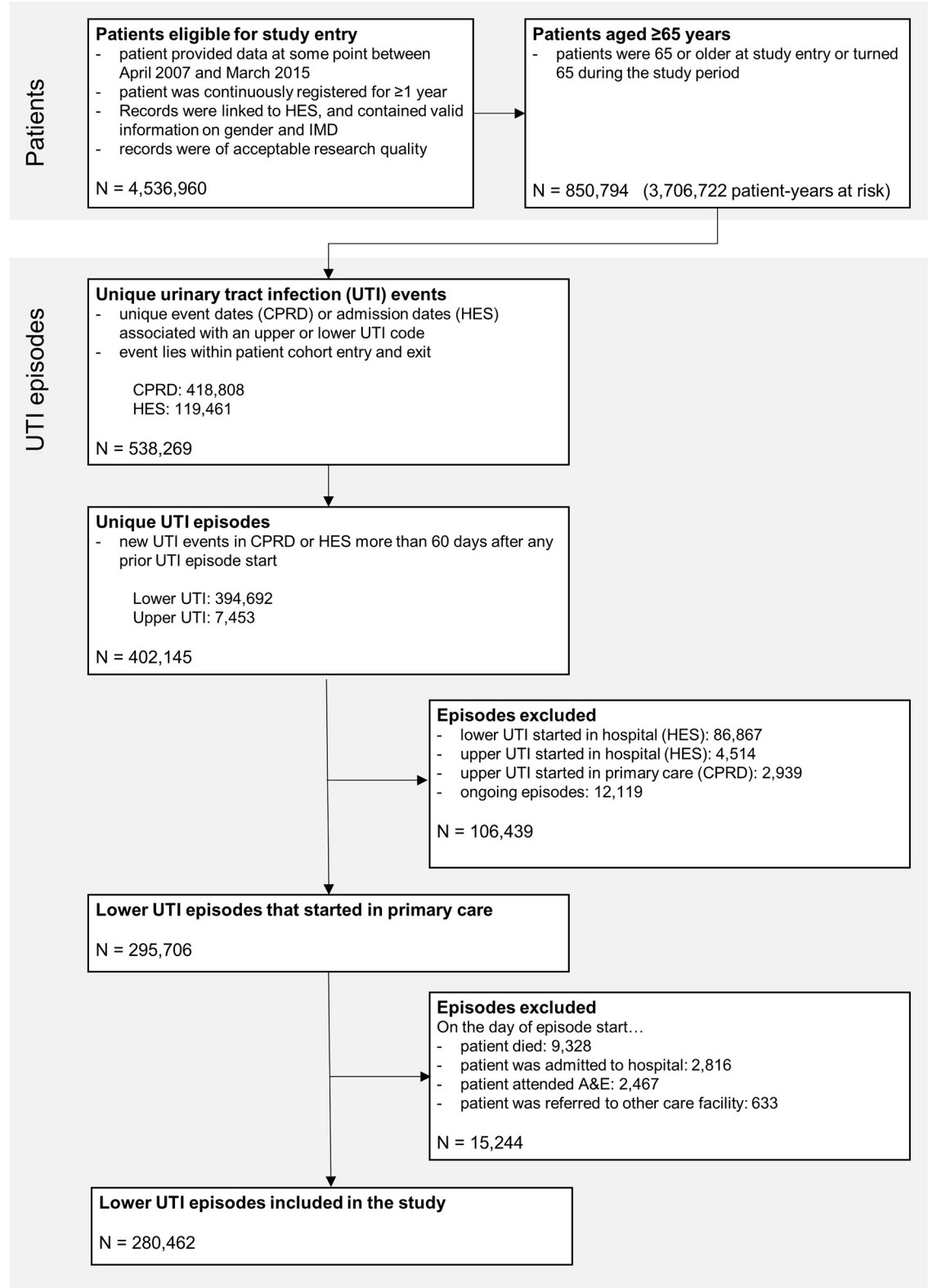

**Fig 2. Selection of the study cohort.** A&E, accident and emergency; CPRD, Clinical Practice Research Datalink; HES, Hospital Episode Statistics; IMD, Index of Multiple Deprivation 2015; UTI, urinary tract infection.

**Table 1. Baseline characteristics associated with lower urinary tract infection episodes in primary care, comparing episodes with and without immediate (same day) antibiotic prescribing.**

| Patient characteristics | All<br>N (%)/median [IQR] | Immediate prescribing<br>N (%)/median [IQR] | No immediate prescription<br>N (%)/median [IQR] | p-value |
|---|---|---|---|---|
| **Total** | 280,462 (100.0) | 244,963 (87.3) | 35,499 (12.7) | |
| **Age** (continuous)* | 77.3 [71.1–83.9] | 77.3 [71.1–83.8]] | 77.8 [71.3–84.7] | <0.001 |
| **Age** (categorical)<br>65–74 | 113,332 (40.4) | 99,511 (40.6) | 13,821 (38.9) | <0.001 |
| 75–84 | 106,900 (38.1) | 93,714 (38.3) | 13,186 (37.1) | |
| ≥85 | 60,230 (21.5) | 51,738 (21.1) | 8,492 (23.9) | |
| **Female** | 217,425 (77.5) | 196,459 (80.2) | 20,966 (59.1) | <0.001 |
| **IMD**<br>Q1 (least deprived) | 69,516 (24.8) | 60,482 (24.7) | 9,034 (25.4) | 0.005 |
| Q2 | 68,320 (24.4) | 59,654 (24.4) | 8,666 (24.4) | |
| Q3 | 62,324 (22.2) | 54,607 (22.3) | 7,717 (21.7) | |
| Q4 | 46,119 (16.4) | 40,404 (16.5) | 5,715 (16.1) | |
| Q5 (most deprived) | 34,183 (12.2) | 29,816 (12.2) | 4,367 (12.3) | |
| **Region**<br>South of England | 116,148 (41.4) | 100,785 (41.1) | 15,363 (43.3) | <0.001 |
| London | 27,066 (9.7) | 23,443 (9.6) | 3,623 (10.2) | |
| Midlands and east of England | 79,274 (28.3) | 69,271 (28.3) | 10,003 (28.2) | |
| North of England and Yorkshire | 57,974 (20.7) | 51,464 (21.0) | 6,510 (18.3) | |
| **NHS financial year**<br>2007/08 | 34,805 (12.4) | 30,928 (12.6) | 3,877 (10.9) | <0.001 |
| 2008/09 | 36,010 (12.8) | 31,936 (13.0) | 4,074 (11.5) | |
| 2009/10 | 36,874 (13.1) | 32,753 (13.4) | 4,121 (11.6) | |
| 2010/11 | 37,159 (13.2) | 32,806 (13.4) | 4,353 (12.3) | |
| 2011/12 | 37,499 (13.4) | 32,652 (13.3) | 4,847 (13.7) | |
| 2012/13 | 37,893 (13.5) | 32,722 (13.4) | 5,171 (14.6) | |
| 2013/14 | 35,386 (12.6) | 30,169 (12.3) | 5,217 (14.7) | |
| 2014/15 | 24,836 (8.9) | 20,997 (8.6) | 3,839 (10.8) | |
| **CCI** (continuous)* | 2 [0–3] | 2 [0–3] | 2 [0–3] | <0.001 |
| **CCI** (categorical)<br>0 | 82,406 (29.4) | 72,475 (29.6) | 9,931 (28.0) | <0.001 |
| ≥1 | 198,056 (70.6) | 172,488 (70.4) | 25,568 (72.0) | |
| **Smoking status**<br>Nonsmoker | 167,927 (59.9) | 147,977 (60.4) | 19,950 (56.2) | <0.001 |
| Ex-smoker | 92,507 (33.0) | 79,419 (32.4) | 13,088 (36.9) | |
| Smoker | 20,028 (7.1) | 17,567 (7.2) | 2,461 (6.9) | |
| **Recurrent UTI** | 71,391 (25.5) | 62,890 (25.7) | 8,501 (23.9) | <0.001 |
| **Hospital stays**<br>Discharged from hospital in prior 7 days | 6,526 (2.3) | 5,396 (2.2) | 1,130 (3.2) | <0.001 |
| Discharged from hospital in prior 30 days | 20,655 (7.4) | 17,682 (7.2) | 2,973 (8.4) | <0.001 |
| Number of days spent in hospital in prior year* | 0 [0–0] | 0 [0–0] | 0 [0–1] | <0.001 |
| Number of admissions in prior year* | 0 [0–0] | 0 [0–0] | 0 [0–0] | <0.001 |
| **A&E attendances**<br>A&E attendance in prior 30 days | 10,875 (3.9) | 8,729 (3.6) | 2,146 (6.0) | <0.001 |
| Number of attendances in prior year* | 0 [0–1] | 0 [0–1] | 0 [0–1] | <0.001 |
| **Antibiotic in prior 30 days** | 54,077 (19.3) | 44,496 (18.2) | 9,581 (27.0) | <0.001 |
| **Index event was home visit** | 12,531 (4.5) | 9,116 (3.7) | 3,415 (9.6) | <0.001 |
| **Outcomes within 60 days after episode start** | | | | |

(*Continued*)

**Table 1.** (Continued)

| Patient characteristics | All | Immediate prescribing | No immediate prescription | |
|---|---|---|---|---|
| | N (%)/median [IQR] | N (%)/median [IQR] | N (%)/median [IQR] | p-value |
| BSI | 1,253 (0.4) | 1,025 (0.4) | 228 (0.6) | <0.001 |
| Days to diagnosis of BSI* | 20 [6.0–39.0] | 22 [7.0–40.0] | 13 [3.0–32.5] | <0.001 |
| Hospitalization (non-BSI, non-UTI) | 16,492 (5.9) | 13,700 (5.6) | 2,792 (7.9) | <0.001 |
| Death | 5,636 (2.0) | 4,593 (1.9) | 1,043 (2.9) | <0.001 |

Note that patients can have more than one UTI episode within the study period and will be counted separately for each of their episodes.

* Coded as a continuous variable. Note that since all continuous variables had a right-skewed distribution they were summarised by median and IQR. A nonparametric Wilcoxon rank-sum test was used to compare differences in continuous variables between groups.

A&E, accident and emergency; BSI, bloodstream infection; CCI, Charlson Comorbidity Index; IMD, Index of Multiple Deprivation 2015; IQR, interquartile range; NHS, UK National Health Service; Q1–Q5, quintiles 1–5; UTI, urinary tract infection.

antibiotics immediately compared with those who were treated immediately (13 days, IQR: 3–32.5 days versus 22 days, IQR: 7–40 days, p-value < 0.001; Table 1).

The crude odds of BSI were higher in patients who were not treated with antibiotics immediately, compared with patients who received a prescription on the date of their first consultation for UTI (OR 1.53, 95% CI 1.33–1.77, p-value < 0.001; Table 2). However, after adjusting for patient demographics, year of consultation, comorbidities, smoking status, recent hospitalizations, recent A&E attendances, recent antibiotic prescribing, and home visits, we found no evidence that delaying or withholding treatment was associated with an increased likelihood of BSI in the following 60 days (adjusted odds ratio [aOR] 1.13, 95% CI 0.97–1.32; p-value = 0.105). The corresponding NNEH was 1,882, i.e., we would anticipate 1 extra case of BSI for every 1,882 patients not treated immediately with antibiotics. The estimated lower bound of the 95% confidence interval was 904, reflecting uncertainty in the OR (upper limit not calculated).

Women were less likely to develop BSI compared with men (aOR 0.49, 95% CI 0.43–0.55, p-value < 0.001; Table 1). Increasing age (aOR 1.22, 95% CI 1.18–1.27 per 5 years, p-value < 0.001) and social deprivation (Q5 versus Q1: aOR 1.45; 95% CI 1.19–1.76, p-value < 0.001) were independently associated with BSI.

Comorbidity, prior hospital admissions, and antibiotic treatment in the prior 30 days were all associated with increased odds of BSI. The odds of BSI were also increased in patients who received a home visit from their GP (aOR 2.19, 95% CI 1.85–2.60, p-value < 0.001), including visits to care homes. We found modest evidence (p = 0.069) that gender, but not age, modified the association between delayed or withheld antibiotics and BSI (Women: aOR 1.27, 95% CI 1.03–1.57, p-value = 0.024; Men: aOR 0.98, 95% CI 0.79–1.21, p-value = 0.845; S2 Table and S3 Table). Because we had not previously hypothesized an interaction between gender and treatment, all subsequent analyses excluded interactions.

Not immediately treating patients with antibiotics was associated with increased mortality in the subsequent 60 days (aOR 1.17, 95% CI 1.09–1.26, p-value < 0.001; S4 Table). The corresponding NNEH was 326, i.e., for every 326 (95% CI 214–641) patients not immediately treated with antibiotics, we observed 1 additional death within 60 days. However, in sensitivity analysis, patients who were not treated immediately with antibiotics were also more likely to have been admitted to hospital for conditions unrelated to BSI or UTI in the 60 days following consultation (aOR 1.20, 95% CI 1.15–1.25, p-value < 0.001; S5 Table). Restricting the analysis to each patient's first episode of UTI supported our main findings of no association between delayed or withheld treatment and BSI (aOR 0.97, 95% CI 0.80–1.19, p-value = 0.774; S6 Table), but shortening the period of follow-up to 30 days provided some evidence of an

**Table 2. Univariable and multivariable associations between immediate antibiotic prescribing for UTI and BSI within 60 days, adjusting for covariates using generalized estimating equations and Huber–White sandwich estimators.**

| Patient characteristics | Univariable analysis | | Multivariable analysis[*] | |
|---|---|---|---|---|
| | OR (95% CI) | *p*-value | aOR (95% CI) | *p*-value |
| **No antibiotic** | 1.53 (1.33–1.77) | <0.001 | 1.13 (0.97–1.32) | 0.105 |
| **Age** (continuous; per 5 years) | 1.32 (1.28–1.36) | <0.001 | 1.22 (1.18–1.27) | <0.001 |
| **Female gender** | 0.40 (0.36–0.45) | <0.001 | 0.49 (0.43–0.55) | <0.001 |
| **IMD** <br> Q1 (least deprived) | 1 | | 1 | |
| Q2 | 1.25 (1.06–1.49) | 0.009 | 1.21 (1.02–1.44) | 0.027 |
| Q3 | 1.29 (1.09–1.54) | 0.004 | 1.21 (1.02–1.44) | 0.028 |
| Q4 | 1.36 (1.14–1.64) | <0.001 | 1.27 (1.06–1.53) | 0.011 |
| Q5 (most deprived) | 1.69 (1.40–2.04) | <0.001 | 1.45 (1.19–1.76) | <0.001 |
| **Region** <br> South of England | 1 | | 1 | |
| London | 1.04 (0.84–1.28) | 0.721 | 1.00 (0.80–1.22) | 0.973 |
| Midlands and East of England | 1.18 (1.03–1.35) | 0.020 | 1.13 (0.98–1.29) | 0.090 |
| North of England and Yorkshire | 1.28 (1.11–1.48) | <0.001 | 1.17 (1.00–1.36) | 0.046 |
| **NHS financial year** <br> 2007/08 | 1 | | 1 | |
| 2008/09 | 0.97 (0.77–1.22) | 0.778 | 0.96 (0.76–1.21) | 0.706 |
| 2009/10 | 0.86 (0.68–1.09) | 0.205 | 0.83 (0.65–1.05) | 0.119 |
| 2010/11 | 1.02 (0.81–1.28) | 0.879 | 0.97 (0.77–1.23) | 0.806 |
| 2011/12 | 0.98 (0.78–1.24) | 0.888 | 0.93 (0.73–1.18) | 0.539 |
| 2012/13 | 1.12 (0.90–1.40) | 0.307 | 1.05 (0.84–1.32) | 0.659 |
| 2013/14 | 1.33 (1.07–1.65) | 0.011 | 1.25 (1.00–1.57) | 0.050 |
| 2014/15 | 1.71 (1.37–2.13) | <0.001 | 1.60 (1.27–2.02) | <0.001 |
| **CCI** (continuous)[†] | 1.88 (1.75–2.02) | <0.001 | 1.41 (1.30–1.52) | <0.001 |
| **Smoking status** <br> Nonsmoker | 1 | | 1 | |
| Ex-smoker | 1.23 (1.09–1.38) | <0.001 | 0.96 (0.85–1.08) | 0.494 |
| Smoker | 1.21 (0.98–1.49) | 0.084 | 1.21 (0.97–1.51) | 0.086 |
| **Recurrent UTI** | 1.01 (0.89–1.15) | 0.857 | | |
| **Hospital stays** <br> Discharged from hospital in prior 7 days | 2.95 (2.35–3.69) | <0.001 | 1.39 (1.04–1.85) | 0.024 |
| Discharged from hospital in prior 30 days | 2.48 (2.13–2.88) | <0.001 | 1.23 (1.00–1.51) | 0.046 |
| Number of days spent in hospital in prior year[†] | 1.22 (1.20–1.24) | <0.001 | 1.08 (1.05–1.11) | <0.001 |
| Number of admissions in prior year[†] | 2.33 (2.16–2.52) | <0.001 | 1.33 (1.13–1.55) | <0.001 |
| **A&E attendances** <br> A&E attendance in prior 30 days | 2.37 (1.94–2.88) | <0.001 | 1.16 (0.91–1.48) | 0.237 |
| Number of attendances in prior year[†] | 1.77 (1.65–1.90) | <0.001 | 0.97 (0.86–1.10) | 0.663 |
| **Antibiotic in prior 30 days** | 1.50 (1.33–1.71) | <0.001 | 1.25 (1.10–1.42) | <0.001 |
| **Index event was home visit** | 3.82 (3.26–4.46) | <0.001 | 2.19 (1.85–2.60) | <0.001 |

A&E, accident and emergency; aOR, adjusted odds ratio; CCI, Charlson Comorbidity Index; IMD, Index of Multiple Deprivation 2015; NHS, UK National Health Service; OR, crude odds ratio; Q1–Q5, quintiles 1–5; UTI, urinary tract infection; 95% CI, 95% confidence interval.

[*]Adjusted for all other variables with *p*-value < 0.2 in the univariable analysis

[†]Transformed using the square root before input into the model. Effect sizes represent the relative change in odds (OR) per 1 unit increase in the square root, that is when the risk factor increases from 0 to 1, from 1 to 4, from 4 to 9, etc. on the original scale.

**Table 3. Healthcare setting and recorded cause of BSI/sepsis* recorded within 60 days of episode start date.**

| | Immediate prescribing | | | No immediate prescription | | |
|---|---|---|---|---|---|---|
| Level of evidence for BSI* | N | % of total | % of setting | N | % of total | % of setting |
| **Total** | 1,025 | 100 | | 228 | 100 | |
| **Hospital-confirmed sepsis** | **716** | **69.9** | **100** | **143** | **62.7** | **100** |
| Urosepsis | 295 | 28.7 | 41.2 | 59 | 25.9 | 41.3 |
| of which primary reason for admission | 105 | 10.2 | 14.7 | 24 | 10.5 | 16.8 |
| Sepsis of other infectious cause | 238 | 23.2 | 33.2 | 59 | 25.9 | 41.3 |
| of which lower respiratory cause | 163 | 15.9 | 22.8 | 37 | 16.2 | 25.9 |
| Unspecified sepsis | 183 | 17.9 | 25.6 | 25 | 11.0 | 17.5 |
| **Sepsis recorded in primary care only** | **309** | **30.1** | **100** | **85** | **37.3** | **100** |
| UTI code in hospital† | 209 | 20.4 | 67.6 | 54 | 23.9 | 63.5 |
| Other infection in hospital | 35 | 3.4 | 11.3 | 9 | 3.9 | 10.6 |
| No infection in hospital | 18 | 1.8 | 5.8 | 4 | 1.8 | 4.7 |
| No record of hospitalization | 47 | 4.6 | 15.2 | 18 | 7.9 | 21.2 |

* Although the terms sepsis and BSI are not interchangeable, ICD-10 diagnostic codes usually record "sepsis" rather than BSI, even in cases with a positive microbial culture of blood. We have therefore interpreted ICD-10 codes for sepsis as evidence of BSI

† In these cases, a diagnosis of lower or upper UTI was recorded as primary or secondary diagnosis in hospital, without any coded hospital reference to sepsis. However, a sepsis diagnosis was recorded for the same day in primary care, likely representing a transcription of the hospital discharge letter into the practice's IT system.

BSI, Bloodstream infection; ICD-10, International Classification of Diseases 10th revision; IT, information technology; UTI, Urinary tract infection

association between delaying/withholding treatment and BSI (aOR 1.26, 95% CI 1.07–1.48, *p*-value = 0.006; S7 Table). Use of propensity scores to address residual confounding led to aORs for the association between delayed/withheld prescribing and BSI that ranged from 1.10 (95% CI 0.95–1.28; *p*-value = 0.209) to 1.27 (95% CI 1.08–1.50, *p*-value = 0.004) depending on the method applied (S8 Table and S9 Table).

Finally, in-depth analysis of the cause of BSI showed that one quarter of cases had urosepsis recorded at some point during hospital admission, with urosepsis listed as the main reason for admission in just 129/1253 (10.3%) of all BSI cases (Table 3). More than one-third of hospital-confirmed BSI cases were attributed to nonurinary sources, mainly respiratory infections. A diagnostic code for BSI was solely recorded in the primary care record in 394 cases (31.4%).

## Discussion

In this study, we did not find evidence of increased risk of BSI in individuals who were not treated immediately with antibiotics (on the date of their initial GP consultation) for suspected UTI. Patients who did not receive antibiotics immediately were more likely to die in the following 60 days, but there was limited evidence that these deaths were attributable to urosepsis. Overall, these findings equate to 1 additional death for every 300 patients aged ≥65 years who were not treated immediately with antibiotics.

Although we found some evidence that individuals who were not prescribed antibiotics immediately were at increased risk of death, these results should be interpreted with some caution because they may be subject to bias and residual confounding. Individuals who were not treated with antibiotics immediately were more likely to be admitted to hospital for reasons unrelated to UTI or BSI, which implies that the risk of BSI and death in some patients included in this study may have been driven by a delay in diagnosing the patient's underlying illness, rather than a delay in initiating antibiotics for lower UTI. This hypothesis is supported by the finding of systematic differences between patients who were or were not treated immediately

with antibiotics across a range of factors (comorbidity, antibiotic use in the prior 30 days) that might influence risk of BSI. Taken together, this provides some evidence that antibiotic prescribing decisions may have been influenced by factors (which cannot be measured in EHRs) that were unrelated to the management of suspected UTI. Furthermore, analysis of diagnostic codes revealed that only half of BSI cases could be linked to a UTI code with clear evidence of a nonurinary source, such as skin or respiratory infection, in more than 25% of cases.

## Comparison with existing literature

Previous studies of alternatives to antibiotics or delayed prescribing for community-onset UTI have usually focused on women aged 18–70 years. A systematic review of trials in young, non-pregnant women reported that antibiotic treatment was associated with more rapid resolution of urinary symptoms and microbiological cure based on urine culture, compared with placebo [24], but not with reduced incidence of pyelonephritis. Delayed prescribing has also been safely used in low-risk women with uncomplicated UTI [25], provided there is adequate safety-netting and self-care advice [2]. For example, in a trial comparing treatments for uncomplicated UTI in women aged 18–65 years [26], women receiving ibuprofen had a higher burden of symptoms but considerably less antibiotic exposure compared with women treated with fosfomycin (incident risk reduction 66.5%, 95% CI 58.8%–74.4%; $p < 0.001$), and two-thirds of patients in the ibuprofen group recovered without antibiotics.

Although the efficacy and safety of delayed prescribing for respiratory tract infections in primary care is well-established [27], implementing similar approaches for UTI is controversial because of concerns around prolongation of symptoms and the potential risk of antimicrobial resistance and complicated UTI resulting from inadequate antibiotic therapy. These issues are particularly relevant in elderly patients who have the highest incidence of community-onset UTI, but also the highest incidence of *E.coli* BSI [3], which may be a consequence of suboptimal antibiotic treatment in primary care. However, it is unclear whether use of delayed prescribing for suspected UTI would be acceptable to GPs or to patients in this age group.

With the exception of Gharbi and colleagues, few studies have evaluated the use of delayed prescribing or alternatives to antibiotics in older adults. These patients arguably have the most to gain from prudent antibiotic prescribing because of their increased risk of adverse outcomes related to antibiotic use [28] and high prevalence of asymptomatic bacteriuria [6]. The major barrier to delaying or withholding antibiotics in these individuals is the risk of UTI-related complications, as reported by Gharbi and colleagues [13]. Our analysis, and concerns raised by other research groups [14], call these findings into question. Recognizing the limitations of analyses based on routine data, we find no evidence of an association between delaying or withholding antibiotics and bloodstream infection but some evidence of increased mortality. The discrepancy between our analysis and that conducted by Gharbi and colleagues is likely to relate to the different approaches used to define community-onset UTI and the limitations of coding in EHRs. In our adjusted analysis, the biggest reduction in effect sizes related to inclusion of information on home visits, possibly because prescribing outside the practice is not recorded electronically, and greater comparability of exposure groups, due to exclusion of cases that did not meet criteria for "community-onset."

## Strengths and limitations of this study

A major strength of our analysis is the use of a large and nationally representative primary care database (>850,000 patients) linked to hospital admissions. This means our estimates can be generalized to the UK population [15]. Linkage of the primary care dataset to HES allowed us

to apply stringent criteria to identify community-onset UTI cases by differentiating new from ongoing UTI episodes and excluding cases that originated in hospital. Sensitivity analyses also support our main conclusions and highlight the limitations of diagnostic coding for BSI.

Limitations relate to the fact that EHRs are designed for clinical care not research. Observational studies using CPRD are at risk of confounding by indication if there are systematic differences (such as the severity of symptoms) between patients who receive a prescription and those who do not. This is particularly challenging when the exposure of interest is unevenly distributed across the study population as seen in this study (87% of patients received an immediate antibiotic versus 13% who did not). Estimates from our propensity score analysis were congruent with our main findings, but we acknowledge that residual confounding is likely. Remaining differences in patient characteristics between groups suggest that we were unable to fully account for confounders. We conclude that residual confounding persists, likely due to factors that influence GPs prescribing decisions but are not well recorded in EHRs, such as severity of clinical presentation; patient's prior medical history; knowledge of patient preferences, for example, in relation to end of life care; and the patient's social circumstances.

Read codes were used to identify patients with suspected UTI, because microbiological culture of urine is usually only performed for patients with recurrent UTI or when the clinician suspects that the patients may have a drug-resistant infection. Consequently, it is likely that our cohort included patients with asymptomatic bacteriuria and/or other types of infections. Up to 40% of prescriptions for nitrofurantoin are not linked to a Read code [23], which suggests that we may have failed to identify some patients who were treated immediately with antibiotics. This also highlights challenges associated with using Read codes to infer the date of infection onset. Similarly, we made the assumption that patients commenced their antibiotic treatment on the date the prescription was issued.

Because delayed prescribing is not well recorded in primary care, it is feasible that some individuals recorded as receiving antibiotics immediately actually received a delayed prescription. Depending on the extent of misclassification between treatment groups, this may have led us to underestimate the reported association between antibiotic treatment and adverse outcomes. However, whereas delayed prescribing is commonly used for respiratory tract infections, evidence to support this approach for UTI is limited. For these reasons, it seems likely that most who were prescribed antibiotics on the date of their consultation started treatment on the same date. Conversely, if there were substantial differences in outcome between delayed prescribing and no prescribing, treating them as 1 group may overestimate the reported association between delayed prescribing and mortality, although the results reported by Gharbi and colleagues [13] suggests that this is not the case. Finally, we used the CCI as a composite measure of comorbidity. This had the advantage of making our analysis comparable with Gharbi and colleagues [13], but it does not take account of the fact that specific comorbidities, such as those affecting the urogenital tract, may impact an individual's risk of BSI more than others (and therefore influence GPs' prescribing decisions).

Cases of sepsis were identified from ICD-10 codes or Read codes, and we found that almost one-third of sepsis diagnoses were only recorded in primary care. It is difficult to disentangle the reasons for this because almost all cases of sepsis are managed in hospital. Patients may have received treatment for sepsis abroad or in a non-NHS setting, or information from the discharge letter may have been used to infer the diagnosis of sepsis. Linkage of microbiological data to HES/CPRD would enable more accurate estimation of the proportion of BSI cases that could be attributed to a urinary source and resolve questions around the proportion of cases with a "sepsis" diagnostic code who have microbiological evidence of BSI.

### Clinical, policy, and research implications

This population-based study highlights uncertainty around the risks and benefits of antibiotic treatment for suspected UTI in patients aged ≥65 years. The increased risk of adverse outcomes in this age-group may make GPs more likely to prescribe antibiotics, increasing the likelihood that these patients will be exposed to antibiotics unnecessarily. For researchers, our findings highlight methodological challenges associated with defining the onset of infection and addressing confounding when analyzing EHRs and the need for linkage of microbiological datasets to HES/CPRD. There is also a need for qualitative research to understand patients' and GPs' views on the acceptability of delayed prescribing for UTI in this age group to inform the design of future studies.

## Conclusion

The safety of delaying or withholding antibiotics in adults aged ≥65 years with suspected UTI is uncertain. Adverse consequences of antibiotic treatment in this population and the public health imperative to tackle antibiotic resistance highlight the need for novel diagnostic and/or risk prediction strategies to guide antibiotic prescribing decisions for suspected UTI.

## Supporting information

**S1 RECORD Checklist. RECORD checklist.** RECORD, REporting of studies Conducted using Observational Routinely-collected Data.
(DOCX)

**S1 Table. Read codes and ICD-10 codes used to define study population, exposures, outcomes and covariates.** ICD-10, International Classification of Diseases 10th revision.
(DOCX)

**S2 Table. Generalized estimating equation models of the association between immediate antibiotic prescribing for UTI and BSI in women.** BSI, bloodstream infection; UTI, urinary tract infection.
(DOCX)

**S3 Table. Generalized estimating equation models of the association between immediate antibiotic prescribing for UTI and BSI in men.** BSI, bloodstream infection; UTI, urinary tract infection.
(DOCX)

**S4 Table. Generalized estimating equation models of the association between immediate antibiotic prescribing for UTI and all-cause mortality within 60 days.** UTI, urinary tract infection.
(DOCX)

**S5 Table. Generalized estimating equation models of the association between immediate antibiotic prescribing for UTI and hospitalization without evidence of UTI or BSI within 60 days.** BSI, bloodstream infection; UTI, urinary tract infection.
(DOCX)

**S6 Table. Generalized estimating equation models of the association between immediate antibiotic prescribing for UTI and BSI within 60 days for a patient's first episode.** BSI, bloodstream infection; UTI, urinary tract infection.
(DOCX)

**S7 Table. Generalized estimating equation models of the association between immediate antibiotic prescribing for UTI and BSI within 30 days among all episodes.** BSI, bloodstream infection; UTI, urinary tract infection.
(DOCX)

**S8 Table. Propensity score analysis (parametric: logistic regression): results of the multivariate analysis are shown for both matching with up to 5 controls and inverse probability weighting.** In the case of matching 2 models were estimated, one using the general estimating equations used in the main analysis and a conditional logistic regression accounting for the matching procedure.
(DOCX)

**S9 Table. Propensity score analysis (nonparametric: gradient boosting machine): results of the multivariate analysis are shown for both matching with up to 5 controls and inverse probability weighting.** In the case of matching 2 models were estimated, one using the general estimating equations used in the main analysis and a conditional logistic regression accounting for the matching procedure.
(DOCX)

**S1 Protocol. The use and protective effect of antibiotics against complications of infection in patients in primary care: a cohort study using linked data from CPRD, HES, and ONS.** CPRD, Clinical Practice Research Datalink; HES, Hospital Episode Statistics; ONS, Office for National Statistics.
(DOC)

**S1 Text. Assertions of type of BSI.** BSI, bloodstream infection.
(DOCX)

## Acknowledgments

The authors are grateful to Dr. Peter Dutey-Magni for helpful comments during the design and execution of the analysis.

This publication presents independent research funded by the National Institute for Health Research (NIHR). The views expressed are those of the authors alone and not necessarily those of the NHS, the NIHR, or the Department of Health and Social Care. This study was carried out as part of the CALIBER program (https://www.ucl.ac.uk/health-informatics/caliber). CALIBER, led from the UCL Institute of Health Informatics, is a research resource consisting of anonymized, coded variables extracted from linked EHRs, methods and tools, specialized infrastructure, and training and support. This study is based in part on data from the CPRD obtained under license from the UK MHRA. The data are provided by patients and collected by the NHS as part of their care and support. This study is further based on data from the HES. Copyright (2019), reused with the permission of The Health & Social Care Information Centre. All rights reserved.

## Author Contributions

**Conceptualization:** Laura Shallcross, Patrick Rockenschaub.

**Formal analysis:** Patrick Rockenschaub.

**Funding acquisition:** Laura Shallcross, Andrew Hayward.

**Methodology:** Laura Shallcross, Patrick Rockenschaub, Nick Freemantle.

**Project administration:** Laura Shallcross.

**Supervision:** Laura Shallcross.

**Validation:** Ruth Blackburn.

**Writing – original draft:** Laura Shallcross, Patrick Rockenschaub.

**Writing – review & editing:** Laura Shallcross, Patrick Rockenschaub, Ruth Blackburn, Irwin Nazareth, Nick Freemantle, Andrew Hayward.

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
