## [Editor Report · Decision Letter 0]

26 Mar 2020

Dear Dr Shallcross, 

Thank you for submitting your manuscript entitled "Antibiotic prescribing for lower respiratory tract infection in older adults in primary care and risk of bloodstream infection: a cohort study using electronic health records" for consideration by PLOS Medicine.

Your manuscript has now been evaluated by the PLOS Medicine editorial staff and I am writing to let you know that we would like to send your submission out for external peer review.

Kind regards,

Caitlin Moyer, Ph.D.,

Associate Editor

PLOS Medicine

---

## [Decision Letter · Decision Letter 1]

4 Jun 2020

Dear Dr. Shallcross,

Thank you very much for submitting your manuscript "Antibiotic prescribing for lower UTI in elderly patients in primary care and risk of bloodstream infection: a cohort study using electronic health records" (PMEDICINE-D-20-00884R1) for consideration at PLOS Medicine. 

[LINK]

In light of these reviews, I am afraid that we will not be able to accept the manuscript for publication in the journal in its current form, but we would like to consider a revised version that addresses the reviewers' and editors' comments. Obviously we cannot make any decision about publication until we have seen the revised manuscript and your response, and we plan to seek re-review by one or more of the reviewers. 

We expect to receive your revised manuscript by Jun 25 2020 11:59PM. Please email us (plosmedicine@plos.org) if you have any questions or concerns.

We look forward to receiving your revised manuscript. 

Sincerely,

Emma Veitch, PhD

PLOS Medicine

On behalf of Clare Stone, PhD, Acting Chief Editor,

PLOS Medicine

plosmedicine.org

*For some reason, the paper title entered as metadata into our submission system mentions antibiotic prescription for lower respiratory tract infection rather than UTI as the exposure variable, which is a bit confusing - if you are able to correct it when submitting the revision, that would be great.

*At this stage, we ask that you include a short, non-technical Author Summary of your research to make findings accessible to a wide audience that includes both scientists and non-scientists. The Author Summary should immediately follow the Abstract in your revised manuscript. This text is subject to editorial change and should be distinct from the scientific abstract. Please see our author guidelines for more information: https://journals.plos.org/plosmedicine/s/revising-your-manuscript#loc-author-summary

*In the last sentence of the Abstract Methods and Findings section, please add a note summarising any key limitation(s) of the study's methodology.

*We'd recommend stating in the Methods paper whether the analytical approach described in this paper corresponded to that laid out in a prospective protocol or analysis plan? Please state this (either way) early in the Methods section.

*The authors might wish to consider using a formal reporting guideline to enhance reporting of their study, before submitting the revision. Choices for this work might include STROBE (for epidemiological studies) or RECORD (for studies involving routinely-collected data):

STROBE - https://www.equator-network.org/reporting-guidelines/strobe/

RECORD - https://www.equator-network.org/reporting-guidelines/record/

It's up to the authors to decide which may be more appropriate, for either, the authors can use the checklist to guide/improve reporting and then append the completed checklist as a supporting information file with the resubmission. 

Comments from the reviewers:

Reviewer #1: This is a well-conducted study on the association between antibiotic prescribing for lower UTI in elderly patients in primary care and risk of bloodstream infection using electronic health records. The study design, datasets, statistical methods and analyses are mostly adequate. Particularly using GEE with a logit link and an exchangeable correlation structure to account for multiple UTI episodes per patient is appropriate. However, there are still a few important issues needing attention.

1) Year of the episode is very important for capturing the time trend/effect on the association. In table 2, it's a bit odd to set the middle year of 2010/11 as benchmark. To be neat and conventional, it would be better to group them by every two years into 4 cohorts and use the first cohort (2007 and 2008) as benchmark for comparison. This will apply for all the analyses in the supplementary tables. Also, to be consistent, the breakdown of year of episode needs to appear in table 1 too.

2) In statistical method on page 10, it says "continuous variables were summarised using means and standard deviations, and categorical variables using absolute numbers and proportions. Wilcoxon rank tests (continuous) and tests (categorical) were used to assess the difference between exposure groups". However, for continuous variables in table 1, it depends on the distribution of the data, for those with normal distribution such as age, they should be summarised using mean and SD and compared with t-test. For those with non-normal (skewed) distribution such as CCI, Number of weeks spent in hospital in prior year, Number of admissions in prior year, and Number of attendances in prior year, they should be summarised using median and IQR and compared using Wilcoxon rank-sum test. In table 1 in the row for Number of attendances in prior year (15,142 (5·4) 0·44 (1·0) 0·53 (1·2) <0·001), the value of 15,142 (5.4) is very strange. Is it a typo? 

3) As pointed out above that a few variables are non-normally distributed, they need to be transformed (maybe using log transformation) into normal variables in the GEE analyses in table 2. This also applies to the other regression analyses in the supplementary tables.

4) As delayed antibiotic prescribing is not well recorded in electronic health records, the authors considered patients who were not prescribed antibiotics and those with a delayed prescription as a single group. This becomes inaccurate and may lead to biased results. The authors need to discuss the impact of this in the limitation and also need to tone down the claims on delayed antibiotic on mortality as the data is imprecise.

Reviewer #2: This a neat and well described study. I have only one major and one or two minor comments

Major comment: there are two relatively recent primary care trials in uncomplicated urinary tract infections (Gagyor et al, your reference 24 and Vik I et al, PLOSMedicine 2018), both showing a clear beneficial effect in women and also less complications (fever, pyelonefritis). So in which elderly patients do we need a placebo controlled trial as you suggest in your discussion? Not in the elderly with a true urinary tract infection I should think? And in those with asymptomatic bacteriuria we do not want a trial either? So, the real challenge is to detect true urinary infection? The authors should discuss this in my opinion.

Minor comments: line 84: the percentage of overprescription depends heavily on the quality of diagnosis and varies across settings and countries, see Butler C et al, Brit J Gen Pract 2017

Line 85: reduction in prescribing was mainly achieved in respiratory infections

Line 204: some comorbidities are much more important than others in terms of risk for complicated infections (for instance cardiac failure versus hip arthritis). Was this taken into account?

An important residual confounder could be that patients did not want to be treated because they had accepted end of life? These elderly patients do not always have a clear higher frailty score than those who full of good spirit

?

Reviewer #3: See attached file (also copied below) 

Peer review UTI

The authors offer a further examination of the issue of so called ‘delayed prescribing’ for adult UTI age >65 and its association with blood stream infection. This is an important topic to guide clinical practice. As the authors point out the diagnosis of UTI in the over 65s is more difficult and there are competing priorities regarding overuse of antibiotics (side effects and resistance) vs potential underuse (risk of complications including sepsis and blood stream infection). An earlier publication [1] suggested a 7-8 fold increase in the risk of BSI following delayed or no treatment. This publication however was controversial due to the definition of UTI episodes and imbalance between the patient characteristics. Plus the serious patient outcomes even in those treated with antibiotics suggests that something unusual was going on with patient selection. Furthermore I would suggest that delayed prescribing as described in the management of RTi is a positive management decision accompanied by information on natural history and safety net advice- usually accompanied by explicit means of access to a prescription. This is quite different to not prescribing but then prescribing later so the original authors[1] use of the term ‘delayed prescribing’ is in itself misleading. Prospective surveys in younger women with UTI show that 93-95%% will receive an antibiotic. [2, 3] Delayed prescribing has not been widely promoted as a management option for UTI although it has been tested in younger female adults in a clinical trial.[4] 

In this paper the authors use the same data source (CPRD) but offer a more rigorous account of case definition. This is illustrated using a Figure (Figure 1) but I have to say I was none the wiser having examined the figure. I think I understand the case definition from the accompanying text but the figure itself was confusing to my mind. Patients with immediate adverse outcomes were also appropriately excluded since their disposition cannot have been influenced by the initial prescribing decision. The authors considered prescription in the following 7 days and non-prescription in the same group, this is appropriate for the reasons above (that this was unlikely a positive management option) and as they correctly point out delayed prescribing is not well coded in primary care records. 

The issue with this data (in common with Ghabi) is that since delayed or non-treatment are not usual strategies for managing lower UTI in younger adults it seems unlikely that it is a usual strategy in a higher risk group (the over 65). Hence it is also probable that those not immediately treated are likely to be different from those offered immediately treatment. This was certainly the case with Ghabi and similarly there are baseline differences evident here. Those with immediate treatment are more likely to be female, are younger, are less deprived, more likely to have a prior history of both admission and antibiotic prescribing and more likely to be visited at home. Some of these differences are marked; female 80% vs 59%, antibiotic exposure 18% vs 27%, home visit 3.7% vs 9.6%. All of these factors were independently associated with risk of BSI so these differences are highly relevant to the analysis. Moreover overall this group has a poor outlook with 6% hospitalised within 60 days and 2% mortality- how does this compare with background rates in the population? In the adjusted analysis non use of antibiotics was not associated with BSI but was with excess mortality risk.

The authors appropriately summarise their findings and consider the limitations of the data which include issues with coding – other studies have demonstrated that many antibiotic prescriptions are not associated with appropriate coding – this analysis can only include those with a code who may differ from those un-coded in terms of symptom severity or certainty of diagnosis. There are also problems with coding of the outcome of interest and one third of sepsis codes were found only in the primary care record and may represent secular changes in coding styles.

The study findings were at odds with those reported by Gharbi but the question of safety of delayed prescribing remains uncertain. I think the authors could make clearer the difference between what is probably being investigated here (no or later prescribing) rather than an active policy to issue a deferred prescription as has been described and trialled for acute respiratory infection. It could be made more explicit that there may be fundamentally different management decisions being enacted due to different population characteristics. Age co-morbidity prior antibiotics and prior admission were all independent risk factors for BSI and are in themselves relevant findings for clinicians.

The authors call for a randomised trial to resolve the uncertainties but with an event rate of only 0.4% this would be a huge trial. Before proposing such a trial we need to better understand the prescribing decisions being made. Does this truly represent a positive strategy are other factors being taken into account. Is the excess mortality risk in those not receiving a prescription actually reflecting underlying risk and part of the decision making process? Is there an appetite in clinicians to offer delayed prescribing in this higher risk group- would they/patients agree to randomisation.

1. Gharbi M, Drysdale JH, Lishman H, Goudie R, Molokhia M, Johnson AP, Holmes AH, Aylin P: Antibiotic management of urinary tract infection in elderly patients in primary care and its association with bloodstream infections and all cause mortality: population based cohort study. BMJ 2019, 364:l525.

2. Little P, Merriman R, Turner S, Rumsby K, Warner G, Lowes J, Smith H, Hawke C, Leydon G, Mullee M et al: Presentation, pattern, and natural course of severe symptoms, and role of antibiotics and antibiotic resistance among patients presenting with suspected uncomplicated urinary tract infection in primary care: observational study. British Medical Journal 2010, 340.

3. Butler CC, Francis N, Thomas-Jones E, Llor C, Bongard E, Moore M, Little P, Bates J, Lau M, Pickles T et al: Variations in presentation, management, and patient outcomes of urinary tract infection: a prospective four-country primary care observational cohort study. Br J Gen Pract 2017.

4. Little P, Moore M, Turner S, Rumsby K, Warner G, Lowes J, Smith H, Hawke C, Leydon G, Arscott A et al: Effectiveness of five different approaches in management of urinary tract infection: randomised controlled trial. British Medical Journal 2010, 340.

[LINK]

---

## [Decision Letter · Decision Letter 2]

28 Jul 2020

Dear Dr. Shallcross,

Thank you very much for re-submitting your manuscript "Antibiotic prescribing for lower UTI in elderly patients in primary care and risk of bloodstream infection: a cohort study using electronic health records" (PMEDICINE-D-20-00884R2) for review by PLOS Medicine.

I have discussed the paper with my colleagues and it was also seen again by two of the original reviewers. Provided that the remaining requests of the reviewers are addressed, and the editorial and production issues are dealt with we are planning to accept the paper for publication in the journal. In particular, please update the presentation of Table 1 data as requested by Reviewer 1, please address the points of Reviewer 3, and please address the Editorial comments, including the adding of p-values to the text of the Results to accompany 95% CIs from your analyses.

[LINK]

We look forward to receiving the revised manuscript by Aug 04 2020 11:59PM. 

Sincerely,

Caitlin Moyer, Ph.D.

Associate Editor 

PLOS Medicine

plosmedicine.org

Requests from Editors:

1.Title: Please mention the study population in the title: “Antibiotic prescribing for lower UTI in elderly patients in primary care and risk of bloodstream infection: a cohort study using electronic health records in England”

2.Abstract: Methods and Findings: For the adjusted odds ratios, please include the important variables that are adjusted for in the analyses.

3.Abstract: Methods and Findings: For associations with BSI or death within 60 days, please report p values in addition to the 95% CIs accompanying the aORs.

4.Abstract: Conclusions: The Conclusions paragraph of the Abstract does not summarize or describe the findings of the study. Please address the study implications without overreaching what can be concluded from the data; the phrase "In this study, we observed ..." may be useful. Please interpret the study based on the results presented in the abstract, emphasizing what is new without overstating your conclusions.

5.Author Summary: “What did the researchers do and find?”: We suggest removing “and undertook extensive supportive analyses” from the first bullet point, as the meaning is vague.

Please reword the second bullet point: “We did not find evidence to suggest that not immediately prescribing antibiotics for UTI increased a patient’s risk of bloodstream infection…”

6.Author Summary: “What did the researchers do and find?”: We suggest changing “likely” to “potential” in the third bullet point, as you cannot determine the extent to which this limitation contributes to your findings.

7.References: Please place the in-text citation in square brackets, before the punctuation mark, like this: [1].

8.Introduction: Page 6, first paragraph: Please temper statements of primary; we suggest: “To the best of our knowledge, Gharbi et al. are the first to…” or similar.

9.Methods: Please add the following statement, or similar, to the Methods: "This study is reported as per the REporting of studies Conducted using Observational Routinely-collected Data (RECORD) guideline (S1 Checklist)."

10.Methods: Please note the nature of participant consent, including whether patient informed consent was written or oral.

11.Results, page 16: Please provide p values in addition to the 95% CIs for the odds ratios of BSI by timing of antibiotic prescription. Please also do this for the adjusted ORs, and mention the factors adjusted for.

12.Results, page 16: Days to BSI diagnosis findings- Please reference the table where these are presented, and please provide p values in the text.

13.Results: Page 16: For the following results, please provide p values in addition to 95% CIs. Please reference the table where the results are shown. “Women were less likely to develop BSI compared to men (OR 0.49, 95% CI: 0.43-0.55). Increasing age (OR 1.22, 95% CI: 1.18-1.27 per 5 years) and social deprivation (Q5 versus Q1: 1.45; 95%-CI: 1.19-1.76) were independently associated with BSI.”

14.Results, throughout: Please provide p values in addition to 95% CIs for all reported analyses.

15.Discussion: First paragraph: Please do not use italics for emphasis in the text. We suggest revising the beginning of the paragraph to: “In this study, we did not find evidence of increased risk of BSI in individuals who were not treated immediately with antibiotics (on the date of their initial GP consultation) for suspected UTI. However, we found that patients who did not receive antibiotics immediately were more likely to die in the following 60 days, but there was limited evidence that these deaths were attributable to urosepsis.” or similar.

16. Discussion: Please present and organize the Discussion as follows: a short, clear summary of the article's findings; what the study adds to existing research and where and why the results may differ from previous research; strengths and limitations of the study; implications and next steps for research, clinical practice, and/or public policy; one-paragraph conclusion.

17.Acknowledgments: Please remove the two copyright symbols from page 25.

18.RECORD checklist: Thank you for including the RECORD checklist. Some of the items have been left blank- please ensure that the checklist is completed.

19.Figure 1: Please define the abbreviation UTI in the legend.

20.Figure 2: Please define abbreviations for HES, IMD, CPRD, and UTI. 

21.Table 1: In the legend, please define abbreviations for UTI, BSI, CCI, OR, and SD.

22.Table 2: In the legend, please list the variables for which you adjusted. Please spell out abbreviations for CCI, UTI, and OR in the legend. 

23.Table 3: In the legend, please define abbreviations for BSI and UTI.

24.Supporting information Table 2: Please provide the unadjusted OR (with 95% CIs and p values) in addition to the adjusted OR. Please note in the legend the variables that were adjusted for in the adjusted analysis. Please define abbreviations for OR, CI, BSI, UTI, NHS in the figure legend.

25.Supporting information Table 3: Please provide the unadjusted OR (with 95% CIs and p values) in addition to the adjusted OR. Please note in the legend the variables that were adjusted for in the adjusted analysis. Please define abbreviations for OR, CI, BSI, UTI, NHS in the figure legend.

26.Supporting information Table 4: Please provide the unadjusted OR (with 95% CIs and p values) in addition to the adjusted OR. Please note in the legend the variables that were adjusted for in the adjusted analysis.Please define abbreviations for OR, CI, BSI, UTI, NHS in the figure legend.

27.Supporting information Table 5: Please define abbreviations for OR, CI, NHS in the figure legend.

28.Supporting information Table 6: Please define abbreviations for OR, CI, NHS in the figure legend.

Comments from Reviewers:

Reviewer #1: Thanks authors for their effort to improve the manuscript. However, I am still not satisfied with the response for comment 1.2 on the summary of data in table 1. The authors' argument doesn't make sense at all. It is very simple rule that normal data should be summarised as mean and SD and non-normal data as median and IQR, and followed by proper t-test or non-parametric test like Wilcoxon rank-sum test. I insist that authors must follow this rule like all the other PLOS Medicine authors to correct this in the paper. By the way, all the other comments were well addressed so it's fine.

Reviewer #3: Thankyou for sharing the revised paper. The authors have done a good job of addressing the concerns of the reviewers.

I think you could add one further bullet to the section what did the reviewers do and find the positive statement reflecting the independent risk factors associated with BSI (below)

Women were less likely to develop BSI compared to men (OR 0.49, 95% CI: 0.43-0.55).

Increasing age (OR 1.22, 95% CI: 1.18-1.27 per 5 years) and social deprivation (Q5 versus

Q1: 1.45; 95%-CI: 1.19-1.76) were independently associated with BSI.

In addition these findings are somewhat supported by a recent publication in BMJ open

Serious bacterial infections and antibiotic prescribing in primary care: cohort study using electronic health records in the UK

https://bmjopen.bmj.com/content/10/2/e036975.abstract

This paper concludes 'We did not find population-level evidence that family practices with lower total antibiotic prescribing might have more frequent occurrence of serious bacterial infections overall.'

[LINK]

---

## [Editor Report · Decision Letter 3]

18 Aug 2020

Dear Dr Shallcross, 

On behalf of my colleagues and the academic editor, Dr. Michael Moore, I am delighted to inform you that your manuscript entitled "Antibiotic prescribing for lower UTI in elderly patients in primary care and risk of bloodstream infection: a cohort study using electronic health records in England" (PMEDICINE-D-20-00884R3) has been accepted for publication in PLOS Medicine. 

PRODUCTION PROCESS

PRESS

PROFILE INFORMATION

Thank you again for submitting the manuscript to PLOS Medicine. We look forward to publishing it. 

Best wishes, 

Caitlin Moyer, Ph.D.

Associate Editor 

PLOS Medicine

plosmedicine.org